# Quantum measurement arrow of time and fluctuation relations for measuring spin of ultracold atoms

Maitreyi Jayaseelan [1,2], Sreenath K. Manikandan [1,2], Andrew N. Jordan[1,2,3] & Nicholas P. Bigelow[1,2 ✉]

The origin of macroscopic irreversibility from microscopically time-reversible dynamical laws—often called the arrow-of-time problem—is of fundamental interest in both science and philosophy. Experimentally probing such questions in quantum theory requires systems with near-perfect isolation from the environment and long coherence times. Ultracold atoms are uniquely suited to this task. We experimentally demonstrate a striking parallel between the statistical irreversibility of wavefunction collapse and the arrow of time problem in the weak measurement of the quantum spin of an atomic cloud. Our experiments include statistically rare events where the arrow of time is inferred backward; nevertheless we provide evidence for absolute irreversibility and a strictly positive average arrow of time for the measurement process, captured by a fluctuation theorem. We further demonstrate absolute irreversibility for measurements performed on a quantum many-body entangled wavefunction—a unique opportunity afforded by our platform—with implications for studying quantum many-body dynamics and quantum thermodynamics.

[1] Department of Physics and Astronomy, University of Rochester, Rochester, NY, USA. [2] Center for Coherence and Quantum Optics, University of Rochester, Rochester, NY, USA. [3] Institute for Quantum Studies, Chapman University, Orange, CA, USA. ✉email: nbig@pas.rochester.edu

The arrow of time, as a mathematical construct, deals with the emergence of irreversibility from time-reversal symmetric dynamical laws[1–10]: the dynamical equations of physics are time-reversal symmetric, while ensembles of physical systems pick out configurations that prefer an increase in entropy, or a preferred arrow of time. Nevertheless, the time symmetry of dynamical equations also implies that if we repeat an experiment sufficiently many times, we might be able to record statistically rare events where the state of the system reverts to its initial conditions. The resolution to the paradoxical situation emerges from the statistics of these experimental realizations: realizations with a backward arrow of time in which the system returns to its initial conditions are exponentially less likely to occur compared to their forward counterparts.

Motivated by this, the inference of a statistical arrow of time can be posed as a game where a physicist is given access to an ensemble of realizations of a particular experiment. For each realization of the experiment, represented by a time-series data, the physicist associates a statistical weight (using Bayesian inference) that quantifies the estimate for how likely it is that the given realization is obtained forward as opposed to backward[7,11]. Such an association arises quite naturally in the context of thermodynamics of small systems[12,13], where a system is considered small in the thermodynamic sense if the energy exchanges are on the order of a few $k_BT$ such that energy fluctuations can be measured[14]; here, the statistical weight for each realization of the experiment corresponds to the entropy produced, and relates to the emergence of laws of thermodynamics for the small system in the form of fluctuation theorems[15,16].

Questions of irreversibility find new relevance today in the context of quantum measurement[11,17–24]. Quantum systems have inherently time-reversal symmetric dynamics. Measurements made on quantum systems, however, are usually described in terms of irreversible wavefunction collapse. Irreversibility in this context seems then to be a question of paramount importance to fundamental quantum theory: is it possible to discuss the emergence of irreversibility in quantum measurements from a time-symmetric measurement dynamics (and how can we quantify this)?

Recent technological advances that allow new classes of measurements on quantum systems, such as quantum weak measurements, enable us to explore this and related questions[25–27]. One of the interesting possibilities that a weak quantum measurement allows is that of undoing a given quantum measurement with some probability, uncollapsing the wavefunction[11,28]. The probability of wavefunction collapse compared to the probability of wavefunction uncollapse can be used to determine an arrow of time for the wavefunction collapse process[7]. This allows us to probe the emergence of irreversibility of quantum measurement in a time-symmetric picture, and its interplay with information acquired in the quantum measurement process[17].

Here, we perform a quantum weak measurement of atomic spin in an ultracold cloud of $^{87}$Rb atoms. 3D ultracold Bose–Einstein condensates (BECs) can be described by a macroscopic wavefunction with long range order, making them a prototypical macroscopic quantum system. Our densities in this experiment are low enough that interatomic interactions do not play a significant role in the system's spin dynamics, so that we may exploit the rigidity of the macroscopic wavefunction of the atomic cloud to realize single-shot ensemble-average measurements in this system. As an additional point of advantage, the identity of the BEC as a macroscopic quantum object supports the entanglement of an independent external degree of freedom, the orbital angular momentum, with the internal spin state, allowing us to extend the realm of quantum fluctuation theorems to systems with quantum many-body interactions. We provide a demonstration of the origin of a statistical arrow of time in the quantum dynamics of ultracold atoms as a result of information acquisition in the quantum measurement process. The excellent isolation from the environment and long coherence times of ultracold atomic systems are necessary for fundamental studies of this nature[29–34]. We also provide experimental evidence to validate the analogous laws of thermodynamics for the arrow of time, formulated in terms of quantum fluctuation theorems.

## Results

**Experiment.** We perform weak measurements of spin in a spinor BEC of $^{87}$Rb prepared in a coherent superposition of spin states using optical Raman imprinting techniques (Fig. 1a and Supplementary Note 1)[35,36]. We begin by considering the case where the system is initialized to a spatially uniform coherent superposition of spin states to provide evidence for absolute irreversibility in quantum spin measurement[17]. The coherent atomic cloud is allowed to fall through a magnetic field gradient $B_0 \approx 300$ G cm$^{-1}$ in the direction of gravity (Fig. 1b). The cloud interacts with the magnetic field for a duration $\tau$ through the magnetic dipole coupling

$$\hat{H}_{SG} = -\boldsymbol{\mu} \cdot \mathbf{B} \tag{1}$$

between the atomic dipole moment $\boldsymbol{\mu}$ and the magnetic field $\mathbf{B}$. The atomic cloud receives a spin-state dependent momentum kick $\delta p_x \approx g_F \mu_B B_0 \tau$ through the unitary evolution

$$\hat{U}_{SG} = \exp\left(-i\delta p_x \hat{\sigma}_z \otimes \hat{x}/\hbar\right), \tag{2}$$

where $g_F$ is the Landé factor and $\mu_B$ the Bohr magneton. Subsequent evolution under free fall for a time of flight $t_f \approx 13$ ms correlates spin state with spatial position. The strength of this correlation (and so the strength of the measurement) is controlled by varying $\tau$ between 500 and 1400 μs. Our readout is then performed with a destructive absorption imaging process that allows us to infer the spin state for each atom in the cloud; a resonant collimated imaging beam propagating along the quantization axis interacts with the atomic cloud and is imaged onto a CCD camera with a simple $4f$ imaging system with unit magnification (Fig. 1c). The matter-based readout scheme using the position of the atom (whose spin is being measured) has the additional advantage that the measurements can be approximated to unit efficiency, unlike photons typically used in quantum optics platforms for readout purposes that are not particle number conserving, leading to inefficiencies in the measurement process from the loss of photons. The ultracold atomic platform is also nearly decoherence free (see Supplementary Note 1), facilitating studies of this nature.

We extend our methods to weak quantum spin measurements performed on a BEC with initially entangled orbital and spin degrees of freedom. The orbital angular momentum is spatially encoded in the atomic spin state using Raman beams carrying orbital angular momentum (see Supplementary Note 1). This serves to demonstrate the versatility of the BEC platform in studying quantum fluctuation theorems in systems with engineered quantum many-body interactions and quantum entanglement.

**Arrow of time.** Here, we discuss the first experiment where a cloud of atoms is prepared in a spin state $|\psi_0\rangle$ that is a coherent superposition of the spin eigenstates $|\psi_\uparrow\rangle$ and $|\psi_\downarrow\rangle$ of the operator $\hat{\sigma}_z$, the projection of atomic spin angular momentum along the quantization axis. Each atom in the spin measurement encodes the information about its arrow of time, which can be inferred from the readout $r$ corresponding to the position of the atom in the cloud absorption image. Using the

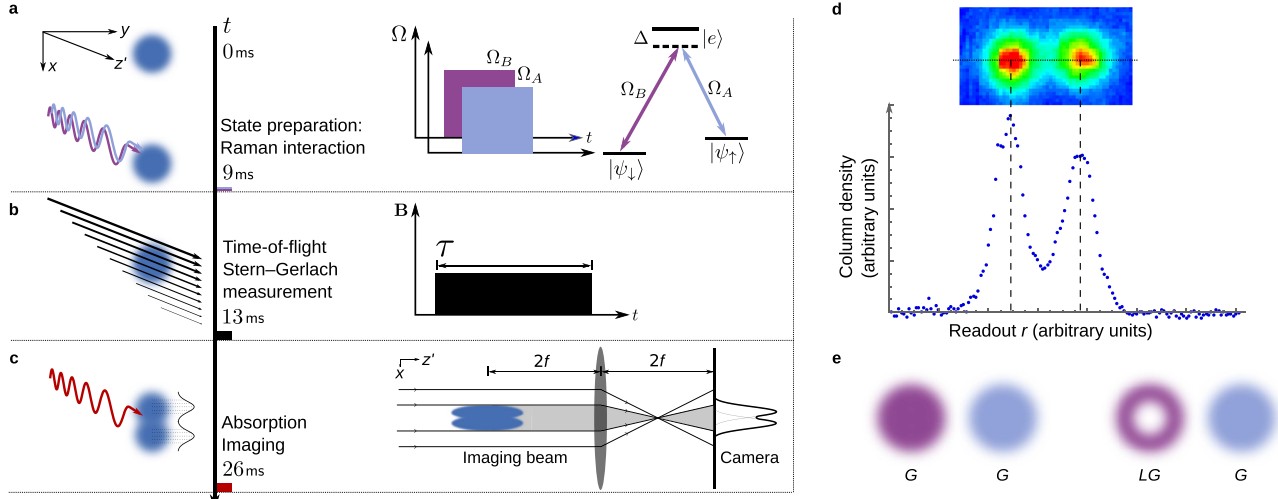

**Fig. 1 Experimental sequence for weakly measuring the quantum spin of an ultracold atomic cloud. a** We prepare the ultracold atomic cloud in a coherent superposition of spin states $\left|\psi_\uparrow\right\rangle$ and $\left|\psi_\downarrow\right\rangle$ with a coherent two-photon Raman interaction using laser fields with Rabi frequencies $\Omega_A$ and $\Omega_B$. **b** The cloud, under free fall through an inhomogeneous magnetic field **B** pulsed on for a time $\tau$, receives a spin-state-dependent momentum kick. The black arrows show the Stern–Gerlach magnetic field that maintains the quantization axis along $z'$ (where we use the primed variable for the lab coordinate $z'$ to distinguish from coordinates on the Bloch sphere), but with a gradient in the direction of gravity, $x$. **c** The spin state is correlated with the readout variable $r$, corresponding to position of atoms in the cloud, after a time of flight $t_f$. Absorption imaging of the ultracold cloud with a 4$f$ imaging system yields the spatially resolved cloud column density (integrated over the $z'$ direction), which provides a single-shot ensemble average of spin measurement. **d** Data are processed by taking a lineout through the center of the cloud that is fit to two Gaussians (for uniform initialization of the cloud). Noise in the imaging process results in negative recorded values for the cloud column density in some pixels (see Supplementary Note 1). These negative values are disregarded in the fit. **e** A uniform initialization of the cloud uses Raman beams with Gaussian (G) modes. The transverse spatial profiles are Gaussian functions. An orbital angular momentum degree of freedom is encoded in the cloud using Raman beams with Laguerre–Gaussian (LG) and G modes. The transverse spatial profile of the LG mode is a donut. This results in an asymmetry between the spatial modes of the spin states in the final absorption image when the cloud is initialized into a spin–orbit entangled state (see below).

quantum Bayesian update rule, the inferred spin state of the atom at location $r$ is[27,37–39],

$$\left|\psi(r)\right\rangle = \frac{\hat{M}(r)\left|\psi_0\right\rangle}{\sqrt{p_F(r)}}. \tag{3}$$

Here $\hat{M}(r)$ is the Gaussian Kraus operator, given by $\hat{M}(r) = \left(2\pi\sigma^2\right)^{-\frac{1}{4}}\exp[-(r-\hat{\sigma}_z)^2/(4\sigma^2)]$, and the forward probability $p_F(r)$ is the probability that an atom ends up at a position corresponding to the readout $r$ in the experiment. The measurement strength is quantified by $\sigma$, the width of the Gaussian clouds in the final post-processed image.

Given the readout variable $r$, we can also envisage a hypothetical backward measurement where the atom is initialized in the final state $\left|\psi(r)\right\rangle$, undergoes a similar spin measurement, and this time ends up with the readout $-r$. The inferred state of the atom in this case is $\propto \hat{M}(-r)\left|\psi(r)\right\rangle \propto \left|\psi_0\right\rangle$. The probability of such a measurement sequence, which would be a realization of a successful measurement reversal, is given by[7,18,40],

$$p_B(r) = ||\hat{M}(-r)\left|\psi(r)\right\rangle||^2 = \frac{|\det\hat{M}(r)|^2}{p_F(r)}, \tag{4}$$

which can be associated to each of the atoms given its position in the absorption image of the forward experiment. A possible physical realization of undoing the measurement using a thin slit and further spin measurements is discussed in Supplementary Note 2.

One may then ask, given the readout $r$, whether the physical evolution of the atom's spin is more likely to be associated to quantum measurement-induced dynamics forward-in-time (wavefunction collapse) or backward-in-time (wavefunction uncollapse), which are time-reversed inverses of each other[7,18].

The likelihood that the atom's position suggests a likely forward evolution can be computed from the Bayesian inference method, yielding a natural discriminator that we call the quantum measurement arrow of time $Q(r)$ for each of the atoms (specified by the readout $r$)[7],

$$Q(r) = \log\frac{p_F(r)}{p_B(r)} = \log\frac{p_F^2(r)}{|\det\hat{M}(r)|^2}. \tag{5}$$

Making the Gaussian approximation to the measurement process, the probability distributions for $Q(r)$ can be constructed from the forward probability distribution $p_F$ and the backward distribution $p_B$ that requires knowledge of the measurement strength, encoded in $\sigma$ (see "Methods"). Note that the arrow of time $Q$, despite being qualitatively similar, has some distinct quantum features compared to the thermodynamic entropy production discussed in similar contexts (see, for instance, ref. [41]). The arrow of time $Q$ is directly linked to exchange of information rather than energy/heat in the quantum measurement process, and diverges when the measurement is strong such that the system is taken towards an eigenstate (measurement becomes more irreversible[7,17]). For the case when the atom's spin is initialized along the median of the Bloch sphere ($z=0$), the probability distribution $p(Q)$ has the following form[7]:

$$p(Q) = \sqrt{\frac{\sigma^2}{2\pi}}\frac{e^Q}{\sqrt{e^Q - 1}}e^{-\frac{1}{2\sigma^2}-\frac{\sigma^2}{2}[\cosh^{-1}(e^{\frac{Q}{2}})]^2}. \tag{6}$$

The distributions have been successfully reconstructed in a quantum trajectory analysis previously[18]. Experimentally constructed probability distributions of the arrow of time are shown in Fig. 2a, c. The green shaded area represents statistically rare events where the arrow of time points backward.

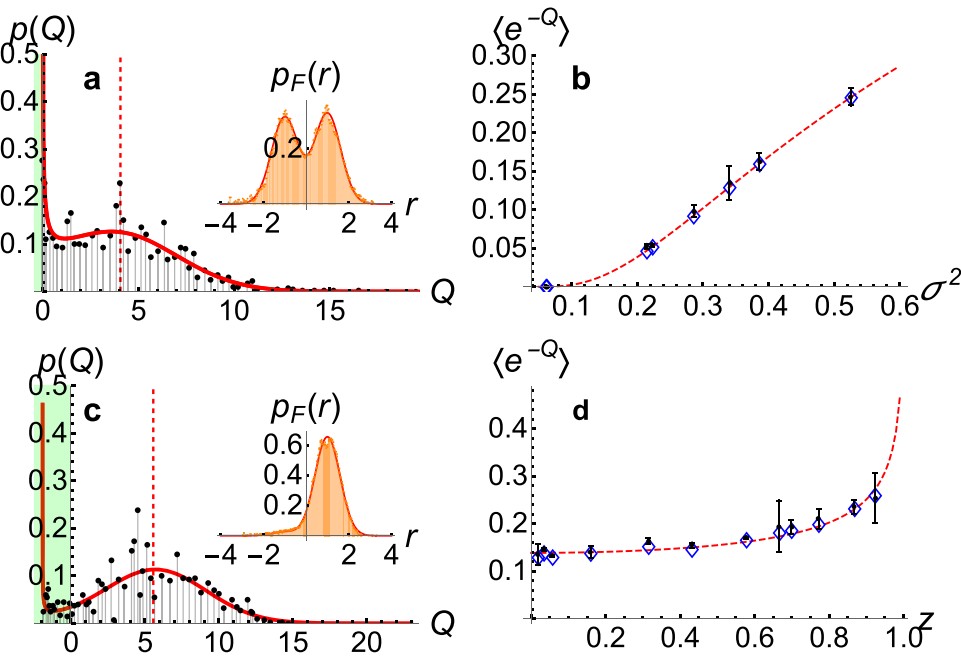

**Fig. 2 Arrow of time distributions and fluctuation relations for uniform initialization of the ultracold cloud. a** Arrow of time distribution, $p(Q)$, for $z = \langle \hat{\sigma}_z \rangle = 0.02$, and $\sigma^2 = 0.39$, where $\sigma$ encodes the measurement strength. The inset shows the corresponding probability distribution, $p_F(r)$, of the readouts, $r$. $p(Q)$ is the distribution of the arrow of time measure $Q$, estimated from the experimental data using Eq. (5) and the form of the measurement operator $\hat{M}(r)$ obtained from the fit to the experimentally determined forward probability distribution. The red profile is the theoretical distribution $p(Q)$ for $z = 0$ [see Eq. (6)]. **b** Absolute irreversibility for $z \approx 0$ for different $\sigma^2$. The black dots with error bars represent prediction for $\langle e^{-Q} \rangle$ by estimating $Q$ from the experimental data using Eq. (5), for single-shot realizations of the experiment. The red dashed line is the theory prediction for $z = 0$. **c** Arrow of time distribution for $z = 0.92$, $\sigma^2 = 0.34$. The negative $Q$ axis is shaded in green, and the red profile indicates the corresponding numerical simulation of $p(Q)$. **d** Absolute irreversibility for different $z$, same $\sigma^2$, for single-shot realizations of the experiment. The black dots with error bars represent prediction for $\langle e^{-Q} \rangle$, the left-hand side of the fluctuation theorem, by estimating $Q$ from the experimental data using Eq. (5). The red dashed line is the theory prediction for $\langle \sigma^2 \rangle = 0.35$. In both **b** and **d**, the center of the blue diamonds (the shape is not intended to provide an error region) indicate the model prediction for $\langle e^{-Q} \rangle \equiv 1 - \mu$[17] obtained by assuming the corresponding fit parameters for each individual realizations in an ideal simulation of the experiment. The error bars shown in **b**, **d** account for systematic errors in the experiment and data processing: the omitted background noise (appearing as negative column densities recorded by our imaging process, see above), the error from renormalizing this column density, and the error in the fit assumed for the experimental data (see Supplementary Note 5).

**Fluctuation theorem.** Fluctuation theorems constrain the probability distribution of the arrow of time $p(Q)$. The integral fluctuation theorem for the arrow of time for weak spin measurements is given by[17,18,42],

$$\langle e^{-Q(r)} \rangle_r = 1 - \mu, \tag{7}$$

where $0 \leq \mu \leq 1$ (see Supplementary Note 3). Here, $\mu$ is the degree of absolute irreversibility in the quantum measurement process, which given the initial state $|\psi_0\rangle$ is defined as[17],

$$\mu = \int dr \, \frac{|\langle \bar{\psi}_0 | \hat{M}(r)^\dagger \hat{M}(r) | \psi_0 \rangle|^2}{p_F(r)}, \tag{8}$$

where $|\bar{\psi}_0\rangle$ is the quantum state orthogonal to $|\psi_0\rangle$, and $\hat{M}(r)^\dagger \hat{M}(r)$ is the effect matrix for the Gaussian spin measurement process. In Eq. (7), we perform the ensemble average over the readout variable $r$, which corresponds to the position of atoms in the cloud. From the convex nature of exponential functions, it immediately follows that the average of the arrow of time has a strict positive lower bound[17,43],

$$\langle Q(r) \rangle_r \geq -\log(1 - \mu). \tag{9}$$

The vertical bars in Fig. 2a, c represent the average arrow of time for each predicted distribution. Equation (9) is analogous to the thermodynamic second law for the quantum measurement process, and suggests that the quantum measurement process has a stronger irreversibility than the usual second law, which only

constrains the first moment of $p(Q)$ by requiring $\langle Q \rangle \geq 0$. Such processes, which have a strong positive lower bound in the statement of second law, are known as absolutely irreversible[17,43]. A canonical example is the free expansion of a single gas particle in a box[43], which is a simple case that generalizes the standard forms of fluctuation theorems in the Jarzynski form[16] or Crooks' form[15] in nonequilibrium thermodynamics, where $\mu = 0$. In the case of quantum measurements, $\mu \to 0$ only when the qubit is initialized in an eigenstate of the measured observable, or when the measurement strength tends to zero. In either of these cases, information acquired from quantum measurements has no effect on the quantum state. Absolute irreversibility (nonzero $\mu$) in quantum measurement is thus associated with the collapse of the quantum state, and occurs as a result of acquiring useful quantum information from measurements. Figure 2b, d show experimental evidence of absolute irreversibility in quantum spin measurements performed on an atomic cloud prepared uniformly in a given spin state, first for the case where measurement strength is varied for a given initial quantum spin state, and then as the initial preparation state is varied while keeping the measurement strength fixed.

While our measurement scheme provides a good estimate for $\langle e^{-Q} \rangle$ demonstrating absolute irreversibility in the quantum measurement process, it does not allow us to independently estimate $\mu$, which would be required to fully verify the quantum fluctuation theorem by experimentally probing both the left- and right-hand sides of Eq. (7). Nevertheless, we estimate $\mu$ in

Supplementary Note 7 for the experimental data based on information available from the theory fit, and discuss why our scheme is more suitable to estimate $\langle e^{-Q}\rangle$.

**Fluctuation theorem with quantum many-body interactions**. We also provide evidence for absolute irreversibility in quantum measurements performed on a system described by a many-body wavefunction that encodes an additional external quantum degree of freedom, a unique opportunity afforded by our cold atom platform. We prepare an atomic wavefunction with coupled spin and orbital degrees of freedom (see Supplementary Note 3), resulting in the following many-body quantum entangled state:

$$|\psi_i\rangle = \alpha|\mathrm{LG}\rangle|\psi'_\uparrow\rangle + \beta|\mathrm{G}\rangle|\psi'_\downarrow\rangle, \qquad (10)$$

where G and LG denote Gaussian and Laguerre–Gaussian spatial modes that encode the orbital angular momentum quantum number $\ell$, and $|\psi'_\uparrow\rangle$ and $|\psi'_\downarrow\rangle$ are the spin states. In Supplementary Note 3, we show that the quantum measurement-induced irreversibility can be computed by mapping this process to a spin measurement of a logical qubit defined by amplitudes $(\alpha, \beta)$. The absolute irreversibility of wavefunction collapse resulting from a Stern–Gerlach absorption imaging measurement for this case can then be quantified by a quantum fluctuation theorem similar to Eq. (7).

In the experiment, the spatial mode of the cloud depends nontrivially on the individual and relative spatially varying attributes of the Raman beams, as well as the time of unitary evolution during the Raman interaction (see Supplementary Note 1). Perfect initialization of the cloud to pure LG and G spatial modes is thus not achieved (however, their orbital quantum numbers are indeed associated with the respective spin states). We empirically associate the following effect matrix, $\hat{M}^\dagger_\mathrm{F}(x,y)\hat{M}_\mathrm{F}(x,y)$, to the measurement process,

$$\hat{M}^\dagger_\mathrm{F}(x,y)\hat{M}_\mathrm{F}(x,y) = \mathrm{diag}\left\{|\mathrm{LG(x,y)}|^2, |\mathrm{H(x,y)}|^2\right\}, \qquad (11)$$

where $|H(x,y)|^2$ is an average of two concentric Gaussian probability densities that fits the experimental data better than a single Gaussian (see Supplementary Note 3). This leads to the fluctuation theorem of the form,

$$\langle e^{-Q(x,y)}\rangle_{x,y} = \int \mathrm{d}x\,\mathrm{d}y\; p_\mathrm{F}(x,y)\; e^{-Q(x,y)} = 1 - \mu. \qquad (12)$$

Figure 3 shows experimental evidence of absolute irreversibility in the application of our theory to this case where the initial preparation of the atomic cloud is to a quantum many-body entangled state. The absolute irreversibility of quantum measurement computed from the experimental data is again seen to increase with measurement strength (increasing relative separation between the spatial-mode entangled spin states). The prediction based on simulation using fit parameters obtained from experimental data is also provided for comparison. The spatial asymmetry between the G and LG spatial modes is reflected in the asymmetry of the fluctuation theorem under the sign flip $z \to -z$ (see Supplementary Note 3). In particular, individual realizations where the position of the atoms in the cloud indicates a larger amplitude for the LG mode are more irreversible (greater $\mu$), as an ideal LG mode has a node at its center where its amplitude vanishes, making the quantum measurement arrow of time diverge at the node.

## Discussion

We now discuss our main results in the paper. First, we performed weak quantum measurements of the spin of $^{87}$Rb atoms in the cloud and obtained the entire measurement statistics for spin measurement of a spin state in a single shot. We

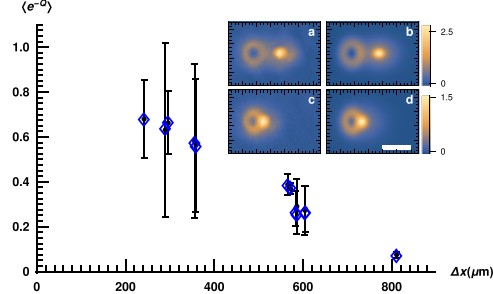

**Fig. 3 Absolute irreversibility of quantum spin measurement for a quantum many-body state.** We quantify how $\langle e^{-Q}\rangle$, the left-hand side of the fluctuation theorem, changes as a function of separation $\Delta x$ between the centers of the Gaussian and Laguerre–Gaussian spatial-mode functions when the BEC is initialized into an entangled quantum state of spin and orbital degrees of freedom (Eq. (10)). The inset shows absorption images from experiment for two datasets (**a** with separation 604.4 μm and **c** with separation 241.9 μm between the centers of the two spatial modes), and the corresponding fits (**b**, **d**) used to compute the theory estimate for $\langle e^{-Q}\rangle$. The white bar in the inset indicates the length scale 640 μm. The color bars indicate column density. Absolute irreversibility increases as stronger measurements are performed (larger the separation). The datasets used here are 12 separate realizations of the experiment with separations $\Delta x$ between the mode centers ranging from 241.89 to 810.92 μm (extracted from the fit), values of $z$ between 0.21 and 0.37, and parameters for the spatial modes as given in Supplementary Figs. 25–36. The centers of the blue diamonds represent the prediction for $\langle e^{-Q}\rangle$ based on the fit parameters in an ideal simulation, while black dots indicate the estimate for $\langle e^{-Q}\rangle$ from the corresponding experimental data. The error bars account for systematic errors in the experiment and data processing (see Supplementary Note 5). The theory values for $\langle e^{-Q}\rangle$ are in good agreement with those estimated from experiment as in the previous case for a uniform initialization of the cloud; however, the error bars are relatively large compared to the previous case, because systematic errors are more pronounced for the current example where we use the whole 2D image in our analysis, while in the previous case (Fig. 2b, d) we only take a single 1D lineout across the image to compute $\langle e^{-Q}\rangle$.

experimentally reconstructed the arrow of time distributions that show remarkable agreement with the corresponding theoretical prediction in Eq. (6). We also considered different initial conditions that produce statistically rare realizations of the quantum measurement process that correspond to cases where the arrow of time points backwards.

We showed that the average of the arrow of time has a strict positive value, and that it increases with the measurement strength. The increasing average arrow of time is a clear signature of acquiring useful quantum information in the measurement process, which causes the irreversible evolution of the atomic spin states towards a spin eigenstate. This feature is captured by the quantum fluctuation theorem[17] as applied to single-shot measurements in our case, for different initializations of the atomic cloud, and for different measurement strengths. In Fig. 2b, d, we demonstrated absolute irreversibility (nonzero $\mu$) for quantum measurements in a uniformly spin initialized cloud of atoms.

Ultracold atomic systems have come to the fore as excellent platforms for simulating systems with complex many-body interactions, making it imperative to understand the role of quantum measurement in these dynamics. We extended the scope of our fluctuation theorem to quantum spin measurements performed on an atomic cloud with engineered quantum many-body interactions through an additional quantum degree of freedom, demonstrating absolute irreversibility for the quantum measurement process in this spin–orbit engineered state

(see Fig. 3). Such states may display dramatically enhanced interactions and a low-energy structure that depends on the spin–orbit interaction, allowing a wide range of interaction strengths to be explored even in quantum systems that are otherwise weakly interacting[44,45]. Our experiment lays the groundwork for studying quantum measurement theory in the context of such exotic many-body wavefunctions where a wealth of physics remains to be explored, including the effect of initial correlations on irreversibility, and the possibility of engineering the low-energy states of the system to tune wavefunction collapse.

## Methods

**Constructing $p(Q)$ from experimental data**. From the experimental data, the numerator of Eq. (5) is estimated by squaring the measured intensity (after normalization), while the denominator is estimated from the experimental data by obtaining the Gaussian fit parameter $\sigma$ for the data, assuming the measurement process is described by the measurement operator $\hat{M}(r)$. This assumption is validated by the fit to the experimental data (see Supplementary Note 4A, B). Now the arrow of time distributions are obtained by converting each $\{Q(r), p_F(r)\} \rightarrow \{Q, p(Q)\}$, by numerically implementing a change of variable $r \rightarrow Q$, multiplying with appropriate Jacobian element for each $Q$. We also verify that this approach leads to the correct probability distribution for $Q$ by simulating the corresponding single qubit measurement process many times to generate the statistics, in Supplementary Note 6.

## Data availability

The data used in this work have been publicly archived in the Zenodo repository at https://doi.org/10.5281/zenodo.4524924.

## Code availability

The Mathematica 10.0 and 12.0 (student edition) notebooks used in the analysis have been publicly archived in the Zenodo repository at https://doi.org/10.5281/zenodo.4524924.

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

## Acknowledgements

We acknowledge helpful conversations with Cyril Elouard, Zekai Chen, Joseph D. Murphree, and S. A. Wadood. N.P.B. and M.J. acknowledge support from the NSF under Grant No. PHY-1708008 and NASA/JPL RSA Grant No. 1616833. A.N.J. and S.K.M. acknowledge support from the NSF under award DMR-1809343.

## Author contributions

M.J. and S.K.M. contributed equally to this work. M.J. collected the experimental data. All authors contributed to analyzing the data and preparing the manuscript. N.P.B. and A.N.J. supervised the collective effort.

## Competing interests

The authors declare no competing interests.
