## [Peer Review File · Nature Communications]

Reviewers' Comments:

Reviewer #1:

Remarks to the Author:

The manuscript reports an experiment with cold atoms to measure the probability distribution of outcomes of a spin weak measurement. From the measurement outcomes, the authors obtain the probability distribution of a variable Q characterizing the likelihood of a forward-in-time process vs a backwards-in-time one and verify an associated fluctuation theorem.

I think that the experiment is a very nice realization of a Stern-Gerlach measurement of atomic spin in a cold atom system. The measurement has high efficiency and gives direct access to the probability of measurement outcomes for long times and without the need to separately follow quantum trajectories.

The one thing I find rather unclear in the paper is how the fluctuation theorem is verified in the experiment (see items 1-3 below). I think it is important to clarify this to appreciate the significance of the results. I have also a further minor question (4) below.

1. If I understand it correctly, $P(Q)$ is not measured directly; what is measured is $P_F(r)$, from which $P(Q)$ is derived using the definition (4). If this is the case, $M(r)$ or $P_B(r)$ should be known. How are they determined (measured independently?, derived from $P_F(r)$...?) Could the authors please clarify?

2. If $P(Q)$ is entirely derived from the measured distribution $P(r)$, the comparison with the theoretical prediction in Fig 2(a-c) and 3(a-c) verifies experimentally that the measurement is well modelled by the gaussian Kraus operator after Eq. (1). Wouldn't this information be completely equivalent to comparing Fig 1(d) with the prediction of the theoretical model? What is the added value of comparing the theory predictions for Q ?

3. It is not quite clear to me how the data in Fig 2 and Fig 3 show the fulfilment of the fluctuation theorem. From Eq. (4) I would expect that one can measure independently the LHS and RHS of the equation in the experiment and show the agreement of data. However, μ in the RHS of Eq. (4) is not defined in terms of measurable quantities. It seems to me that Fig. 2(d) shows again the agreement of the values of $\langle e^{-Q} \rangle$ and the predictions of the theoretical model for that quantity, but I don't see how this proves the fluctuation theorem.

4. Measurements of different strength have been obtained by tuning the time τ . I expect that for longer times some decoherence effects come into play. What are the typical times for which the system and quantum meter can be regarded as coherent? What would decoherence effect imply for the arrow of time fluctuation theorem?

Overall, if the authors can clarify how their data verify the fluctuation theorem, I think the paper is a significant step forward and can open new ways to use cold atoms for quantum measurement experiments.

Reviewer #2:

Remarks to the Author:

In their manuscript Jayaseelan et al provide an experimental measurement of entropy production and fluctuation theorem for a weakly monitored atomic cloud. The study follows a recent theoretical proposal of Prof. Jordan, where weak measurements are characterized as sources of irreversibility using the tools of stochastic thermodynamics.

The experiment is realized with a condensate of Rubidium atoms, the measured observable being the atomic spin. The measurement is based on a Time-of-Flight and its strength can be varied. The authors have measured the stochastic entropy production for each trajectory, that is labelled by the continuous measurement outcome r . These data allow them to plot the histograms of entropy production and the subsequent fluctuation theorem for various measurement strengths and initial spin states. They evidence the presence of absolute irreversibility, which is characteristic of quantum measurement as soon as wave function collapse takes place.

The experiment is convincing, the paper is extremely well written and the agreement with the theory is stunning. To my knowledge this is the first experiment that evidences the quantum measurement induced irreversibility. There have been a few experiments with many body systems dedicated to measure entropy production, but time arrow was emerging from the coupling to a thermal bath (Ref. [17]).

The only questions that I have are related to the positioning/wording:

1) What is the advantage of using a many body system here? These measurements could be performed on a single superconducting qubit where many proposals of quantum thermodynamics have already been probed.

I would naturally expect that many body systems are used to investigate the thermodynamic limit, and that it opens to different physics like phase transitions. This is a big challenge for quantum thermodynamics, that needs to re-define this thermodynamic limit, especially when sources of irreversibility are not the usual thermal ones.

But in the specific case of this paper, I have the impression that the only advantage of using a many body system is that it allows to get the desired statistics in one time of flight, instead of having to repeat many times the same experimental sequence as it would be the case for a superconducting qubit. The advantage is thus more practical - which is a bit frustrating since there should be much more physics to extract!

2) In the same spirit, it would be useful that the authors spend a bit more room to compare their results with those of the Ref [17], which is the other experiment where entropy production was measured in a many body system.

3) I notice that the authors avoid using the word "stochastic entropy production". Actually, $Q(r)$ does not have a real name - while the "arrow of time" is the title of the paper. Why is that so? If it is different than an entropy production, it would be nice to elaborate why.

I guess it has to do with the source of irreversibility, which is quantum measurement and not thermal noise (where the term entropy production was introduced). If it is the case, it would be important to comment the choice of wording. In other words, I encourage the authors to commit and elaborate a bit more on concepts, because their results very nicely illustrate current debates in the community of quantum thermodynamics.

When these questions are answered, I strongly recommend publication.

Reviewer #3:

Remarks to the Author:

The authors present a rather beautiful experiment where a cloud of ultracold atoms is sourced from a Bose-Einstein condensate and is prepared in a coherent superposition of spin states. As the spin states are normally indistinguishable on absorption imaging, a brief pulse of a magnetic gradient field provides a spin-dependent momentum kick to the atoms. This then correlates the

atom position with its spin state, resulting in a weak measurement where the strength of the measurement is controlled by the time over which the gradient field is applied.

In realising the weak measurement of spin on an ensemble of many particles, the authors are able to apply the analysis of Refs. [7,20] which describe the emergence of a statistical arrow of time for a continuously monitored quantum system. The ambiguity in the direction of time evolution during the weak measurement is obtained from two-mode Gaussian fits to the data, which permit the construction of the discriminator Q from Ref.[7], which for some of the experimental data points realises results that are more consistent with backwards evolution in time – that the wavefunction is “uncollapsed” by the weak measurement.

The results of the paper are straightforward, although review of Ref. [7] was useful, and the use of an ensemble of atoms permits the reconstruction of the ensemble measurement and analysis of the arrow of time from a single measurement, rather than many trajectories. I find the results quite remarkable and appreciated the paper’s introduction and approach to the importance of the arrow-of-time question in physics.

However, the results of the theory proposal (Ref. [7]) have already been applied to an experiment with transmon microwave qubits, Ref. [19], through repeated measurement and reconstructions of quantum trajectories. This method was suggested in the original theory proposal, Ref. [7]. It is difficult to see what the current manuscript has to add to this story other than that the experiments can be performed in a single shot. The paper makes no attempts to distinguish the novelty of the reported results from Ref. [19].

The paper suggests that complex many-body interactions justify a study of the arrow of time in this system – however, the experimental preparation relies on the atoms being non-interacting after a time of flight, before the spin superposition is created. The experiment is thus best thought of as N -times single particle experiments.

Given these considerations, at this stage I find the paper inappropriate for Nature Communications, even though the subject matter is clearly of wide interest, the experimental results and analysis quite beautiful and carefully performed, and the presentation of the paper well executed. The paper would be more suitable for a discipline-specific journal, such as Communications Physics. I would find the paper appropriate for Nature Communications if the novelty with respect to Ref. [19] was better established.

There are some suggested improvements to the text and supplemental materials detailed below:

1. I found it unclear how the forward and backward probabilities are determined experimentally from the data. The current details in the Supplemental are unclear, in particular how the backward probability P_B is determined from the experimental data. This should be expanded in the Supplemental.
2. Figure fonts are far too small throughout – the axes of Figs. 2 and 3 are too small to read, similarly the figures in the Supplemental have the same issue.
3. The ordering of the figures in the Supplemental should be adjusted. It was difficult to read with the figures splitting up the text, it perhaps would be better to move all the figures for Sections II and III to after the text.
4. It is unnecessary to explain the absorption imaging techniques in the Supplemental – these are standard techniques and could be replaced by reference to any number of relevant papers.

REVIEWER COMMENTS

Reviewer #1 (Remarks to the Author):

The manuscript reports an experiment with cold atoms to measure the probability distribution of outcomes of a spin weak measurement. From the measurement outcomes, the authors obtain the probability distribution of a variable Q characterizing the likelihood of a forward-in-time process vs a backwards-in-time one and verify an associated fluctuation theorem.

I think that the experiment is a very nice realization of a Stern-Gerlach measurement of atomic spin in a cold atom system. The measurement has high efficiency and gives direct access to the probability of measurement outcomes for long times and without the need to separately follow quantum trajectories.

The one thing I find rather unclear in the paper is how the fluctuation theorem is verified in the experiment (see items 1-3 below). I think it is important to clarify this to appreciate the significance of the results. I have also a further minor question (4) below.

1. If I understand it correctly, $P(Q)$ is not measured directly; what is measured is $P_F(r)$, from which $P(Q)$ is derived using the definition (4). If this is the case, $M(r)$ or $P_B(r)$ should be known. How are they determined (measured independently?, derived from $P_F(r)$...?) Could the authors please clarify?

We have added a brief discussion in the Methods section, as well as a detailed discussion in the supplemental material to clarify this. $P(Q)$ is derived from the measured distribution $P(r)$. This requires estimating $Q(r)$ from the experimental data using eq. 2 of the main text, and then implementing a change of variable numerically. Directly probing $P_B(r)$ is not possible in our platform. To validate our approach, we include an example where the arrow of time statistics is generated by simulating an experiment where a qubit is repeatedly measured to generate the statistics (see supplemental materials). We show that both approaches agree.

2. If $P(Q)$ is entirely derived from the measured distribution $P(r)$, the comparison with the theoretical prediction in Fig 2(a-c) and 3(a-c) verifies experimentally that the measurement is well modeled by the Gaussian Kraus operator after Eq. (1). Wouldn't this information be completely equivalent to comparing Fig 1(d) with the prediction of the theoretical model? What is the added value of comparing the theory predictions for Q ?

While the theory fit is completely equivalent whether done at the level of $P_F(r)$ or $P(Q)$ derived from $P_F(r)$, the distribution $P(Q)$ contains useful information

about the measurement irreversibility and the arrow of time which is not evident from $P_F(r)$. These distributions are analogous to the probability distribution of entropy production in stochastic thermodynamics but from an information theoretic approach. For instance, $P(Q)$ distributions show that the average arrow of time is positive, which is analogous to the second law of thermodynamics in the measurement context. This conclusion is not so evident from $P_F(r)$. Further, estimating the LHS of the fluctuation theorem from the experimental data gives a strong lower bound for the second law, which contains information from all the higher order moments of the probability distribution of Q as well. This feature is also further explained in the SM.

3. It is not quite clear to me how the data in Fig 2 and Fig 3 show the fulfilment of the fluctuation theorem. From Eq. (4) I would expect that one can measure independently the LHS and RHS of the equation in the experiment and show the agreement of data. However, μ in the RHS of Eq. (4) is not defined in terms of measurable quantities. It seems to me that Fig. 2(d) shows again the agreement of the values of $\langle e^{-Q} \rangle$ and the predictions of the theoretical model for that quantity, but I don't see how this proves the fluctuation theorem.

In response to address the referee's comment, we have done additional analysis of the RHS of the FT for the experimental data from our first submission and have included a detailed analysis to the supplemental materials. Our experiment gives a more accurate prediction for the LHS of the FT from the experimental data; in the supplemental materials we discuss why our scheme is more suitable to estimate the LHS, while also providing an estimate based the experimental data for the RHS. We have rephrased sentences to clarify this point and have added a paragraph to emphasize that our analysis in the main text provides evidence for absolute irreversibility in the measurement process, based on probing the LHS of the fluctuation theorem.

4. Measurements of different strength have been obtained by tuning the time τ . I expect that for longer times some decoherence effects come into play. What are the typical times for which the system and quantum meter can be regarded as coherent? What would decoherence effect imply for the arrow of time fluctuation theorem?

The ultracold temperature and isolation of the atomic cloud from the environment allow us to treat our atomic spin state as coherent over the timescales of the experiment. In the supplemental materials, we have added additional data showing interference of spatial modes of the cloud to demonstrate coherence at the time of absorption imaging and for a measurement time of 1100 microseconds. The spatial and spin coherence is shown to be maintained on the timescales of our experiment. It is not possible to probe longer timescales with our current setup, since the cloud under free-fall leaves the field of view of the camera and falls to the bottom of the vacuum

chamber within 30 ms after release from the trap (absorption images presented are taken at 26 ms after release). Decoherence effects in BEC experiments are typically extremely well controlled, so that the coherence time of trapped atomic clouds is on the order of minutes. For measurements taking more time than the coherence time, decoherence effects come into play, and we would need to employ a version of fluctuation theorem for initial mixed states, presented in the appendix of Ref. [17].

Overall, if the authors can clarify how their data verify the fluctuation theorem, I think the paper is a significant step forward and can open new ways to use cold atoms for quantum measurement experiments.

We thank the referee for the very helpful comments.

Reviewer #2 (Remarks to the Author):

In their manuscript Jayaseelan et al provide an experimental measurement of entropy production and fluctuation theorem for a weakly monitored atomic cloud. The study follows a recent theoretical proposal of Prof. Jordan, where weak measurements are characterized as sources of irreversibility using the tools of stochastic thermodynamics.

The experiment is realized with a condensate of Rubidium atoms, the measured observable being the atomic spin. The measurement is based on a Time-of-Flight and its strength can be varied. The authors have measured the stochastic entropy production for each trajectory, that is labelled by the continuous measurement outcome r . These data allow them to plot the histograms of entropy production and the subsequent fluctuation theorem for various measurement strengths and initial spin states. They evidence the presence of absolute irreversibility, which is characteristic of quantum measurement as soon as wave function collapse takes place.

The experiment is convincing, the paper is extremely well written and the agreement with the theory is stunning. To my knowledge this is the first experiment that evidences the quantum measurement induced irreversibility. There have been a few experiments with many body systems dedicated to measure entropy production, but time arrow was emerging from the coupling to a thermal bath (Ref. [17]).

The only questions that I have are related to the positioning/wording:

1) What is the advantage of using a many body system here? These measurements could be performed on a single superconducting qubit where many proposals of quantum thermodynamics have already been probed.

But in the specific case of this paper, I have the impression that the only advantage of using a many body system is that it allows to get the desired statistics in one time of flight, instead of having to repeat many times the same experimental sequence as it would be the case for a superconducting qubit. The advantage is thus more practical - which is a bit frustrating since there should be much more physics to extract!

I would naturally expect that many body systems are used to investigate the thermodynamic limit, and that it opens to different physics like phase transitions. This is a big challenge for quantum thermodynamics, that needs to re-define this thermodynamic limit, especially when sources of irreversibility are not the usual thermal ones.

We thank the referee for motivating us to perform further studies using our cold atom system. We performed new experiments to probe the unique advantage of ultracold atoms to address the referee's comment. We have included arrow of time and fluctuation theorem analyses for spin measurements on an atomic cloud with entanglement between multiple quantum degrees of freedom. We have also provided additional theory analysis for the new experiment in the supplemental materials. The experiment demonstrates the unique advantage of ultracold atoms in investigating quantum fluctuation theorems in contexts where more complex quantum many-body dynamics is present. The "spin-orbit coupled" quantum state used in the new experiment has garnered tremendous interest in the quantum simulation community as it affords the opportunity to tune interactions even in systems that are otherwise weakly interacting, for instance; our new experiment is a first step in understanding the role of measurement in studies performed on these exotic states.

An additional advantage we want to emphasize is the close to unit efficiency as opposed to the superconducting qubit platforms (see ref. 18 of main text) where the readout efficiency is much smaller, and therefore additional modeling to account for this inefficiency is required to discuss the arrow of time.

2) In the same spirit, it would be useful that the authors spend a bit more room to compare their results with those of the Ref [17], which is the other experiment where entropy production was measured in a many body system.

We have done some comparison to this work as suggested by the referee.

3) I notice that the authors avoid using the word "stochastic entropy production". Actually, $Q(r)$ does not have a real name - while the "arrow of time" is the title of the paper. Why is that so? If it is different than an entropy

production, it would be nice to elaborate why.

I guess it has to do with the source of irreversibility, which is quantum measurement and not thermal noise (where the term entropy production was introduced). If it is the case, it would be important to comment the choice of wording. In other words, I encourage the authors to commit and elaborate a bit more on concepts, because their results very nicely illustrate current debates in the community of quantum thermodynamics.

Yes, $Q(r)$ is the arrow of time we are referring to, which is now clarified in the paper. We avoid using the word entropy production as that typically relates to thermal systems where there is entropy/heat exchange with an environment. In our case, instead of exchanging entropy/heat, information is exchanged as a result of the quantum weak measurement process. We have added sentences in the main text to emphasize this difference.

When these questions are answered, I strongly recommend publication.

We thank the referee for their positive recommendation of publication of our article in Nature Communications.

Reviewer #3 (Remarks to the Author):

The authors present a rather beautiful experiment where a cloud of ultracold atoms is sourced from a Bose-Einstein condensate and is prepared in a coherent superposition of spin states. As the spin states are normally indistinguishable on absorption imaging, a brief pulse of a magnetic gradient field provides a spin-dependent momentum kick to the atoms. This then correlates the atom position with its spin state, resulting in a weak measurement where the strength of the measurement is controlled by the time over which the gradient field is applied.

In realising the weak measurement of spin on an ensemble of many particles, the authors are able to apply the analysis of Refs. [7,20] which describe the emergence of a statistical arrow of time for a continuously monitored quantum system. The ambiguity in the direction of time evolution during the weak measurement is obtained from two-mode Gaussian fits to the data, which permit the construction of the discriminator Q from Ref.[7], which for some of the experimental data points realises results that are more consistent with backwards evolution in time – that the wavefunction is “uncollapsed” by the weak measurement.

The results of the paper are straightforward, although review of Ref. [7] was useful, and the use of an ensemble of atoms permits the reconstruction of the ensemble measurement and analysis of the arrow of time from a single

measurement, rather than many trajectories. I find the results quite remarkable and appreciated the paper's introduction and approach to the importance of the arrow-of-time question in physics.

However, the results of the theory proposal (Ref. [7]) have already been applied to an experiment with transmon microwave qubits, Ref. [19], through repeated measurement and reconstructions of quantum trajectories. This method was suggested in the original theory proposal, Ref. [7]. It is difficult to see what the current manuscript has to add to this story other than that the experiments can be performed in a single shot. The paper makes no attempts to distinguish the novelty of the reported results from Ref. [19].

The paper suggests that complex many-body interactions justify a study of the arrow of time in this system – however, the experimental preparation relies on the atoms being non-interacting after a time of flight, before the spin superposition is created. The experiment is thus best thought of as N-times single particle experiments.

Given these considerations, at this stage I find the paper inappropriate for Nature Communications, even though the subject matter is clearly of wide interest, the experimental results and analysis quite beautiful and carefully performed, and the presentation of the paper well executed. The paper would be more suitable for a discipline-specific journal, such as Communications Physics. I would find the paper appropriate for Nature Communications if the novelty with respect to Ref. [19] was better established.

We have added new data from an additional experiment that we performed, where quantum many body interactions are present in the initial conditions of the atomic cloud resulting in quantum entanglement of multiple quantum degrees of freedom. This is a unique advantage ultracold atoms offer compared to superconducting qubits. Additionally, we point out that the absorption imaging process has close to unit efficiency in detecting atoms in the cloud, making them an ideal platform to perform such studies in addition to single shot realizations of the entire statistics where the position of atoms in the cloud is measured.

There are some suggested improvements to the text and supplemental materials detailed below:

1. I found it unclear how the forward and backward probabilities are determined experimentally from the data. The current details in the Supplemental are unclear, in particular how the backward probability PB is determined from the experimental data. This should be expanded in the Supplemental.

The forward probability is directly measured, while the backward probability is inferred using the forward probability and the model. We have improved the main text, methods and the supplemental material to clarify this.

2. Figure fonts are far too small throughout – the axes of Figs. 2 and 3 are too small to read, similarly the figures in the Supplemental have the same issue.
Improved.

3. The ordering of the figures in the Supplemental should be adjusted. It was difficult to read with the figures splitting up the text, it perhaps would be better to move all the figures for Sections II and III to after the text.

Improved

4. It is unnecessary to explain the absorption imaging techniques in the Supplemental – these are standard techniques and could be replaced by reference to any number of relevant papers.

Thanks, changed accordingly.

Summary of changes made:

- 1. New experimental data added with spin dependent LG-G mode separation of the atomic cloud. The main text changed significantly to reflect this, and new sections are added in the SM. The experimental state preparation section in the SM was changed to accommodate the further description necessary.*
- 2. Images added to the supplemental materials demonstrating interference of spatial modes of the atomic cloud, to show persistent coherence within the timescales relevant to the experiment we report.*
- 3. Title is changed slightly to “Quantum measurement arrow of time and fluctuation relations for measuring spin of ultracold atoms”*
- 4. Figures have been replaced in view of the new experiment we did.*
- 5. We have included additional systematic errors in the analysis (with a new discussion in the Supplemental Materials pertaining to our error analysis) and therefore now resort to mentioning the one-sigma error bars arising from experimental imperfections and image processing, replacing the previous confidence intervals based on variance of the sample mean. The previous confidence intervals were directly estimated as σ/\sqrt{N} , where σ^2 was the variance of $\exp(-Q)$ and N the number of pixels used. They corresponded to $\pm \sqrt{n}\sigma$ confidence intervals where n is the average number of counts per pixel. We realized that this approach did not account for some of the systematic errors, which are more appropriate for the single shot imaging scheme we use. This revision was also motivated by the new experiment we performed with LG-G modes*

*and quantum many-body interactions in the initial conditions of the cloud. The systematic errors we now include in the analysis are more pronounced in the new LG-G experiment when the whole 2D image is used in the analysis (whereas a single linecut was used in the G-G experiment). For such experiments with single shot imaging having a large yield, the total number of counts $M(=n*N$ where N is the number of pixels) is very large (based on photon count conversion factors we roughly estimate $n\sim 10^5$, yielding $M\sim 10^9$ number of photons per image assuming $100*100$ pixels in a 2D cropped image) such that the one-sigma standard deviation of the sample mean is smaller by at least one or two orders of magnitude compared to the other one-sigma systematic errors, and is neglected.*

- 6. We have included a comparison between LHS and RHS of the fluctuation theorem for the experimental data in the SM, where the above-mentioned error bars arising from systematic errors are included.*

Reviewers' Comments:

Reviewer #3:

Remarks to the Author:

Please see attached comments and recommendation.

Reviewer #4:

Remarks to the Author:

In this work Jayaseelan et al study experimentally, the statistical irreversibility of weak quantum measurements in a quantum spin of an atom cloud. The theoretical background for this which was developed in previous works by one of the authors of this paper and collaborators (Andrew Jordan) and I invested some time to read them before submitting this review. In essence - if you look at the read out of some measurement scheme the idea is that given the readout - you can tell if the evolution of the spin is likely be due to forward in time or backward in time corresponding to collapse and uncollapse respectively.

Previous theory works have defined a stochastic variable and it has been shown to fulfil a type of fluctuation relation which is interesting because in principle the integral of the exponentiated quantity over the distribution can in principle quantify the strength of the measurement. Unfortunately this experiment does not allow for an independent estimate of this strength but nevertheless does allow for a good estimated for the exponentiated quantity. I mean I have several issues, mainly semantical, regarding the definitions here but whichever way you look at this - it is a wonderful experiment that is an important step towards experimentally quantifying the irreversibility of the quantum measurement process. Although structurally there are strong analogies with results in stochastic thermodynamics - I feel the strongest and most interesting part of the work is that there is some operational notion of irreversibility of a quantum measurement which can be extracted in a very very beautiful and simple cold atoms experiment.

The authors have made a great effort to deal with the previous referees comments and I think now after reading the entire file that this work deserves publication in Nature Communications.

Comments on revision of Nature Communications NCOMMS-20-18641A

I would like to thank the authors for their thorough responses to the previous comments and the paper is much improved. It is also quite interesting to see the additional included data. I believe this furthermore strengthens the paper, while highlighting more sufficiently the specific advantages of undertaking this experiment with cold atoms as opposed to the microwave qubit systems. I'm happy to recommend publication with rectification of the following minor points:

- In Figure 2, it would be useful to label '0' on the horizontal axes and also increase the tick mark line widths to be more visible, as it isn't immediately obvious from the plots that $p(\mathcal{Q})$ ranges to negative values. There is also a font mismatch between the axes labels and the latex formatted figure caption and text.
- Also in Fig. 2, and the caption of Fig. 2, there is a lower case 'p' used for $P(\mathcal{Q})$ which also mismatches the use in the text.
- It appeared to me that the lack of a definition of μ around Eqn. 4 is a bit frustrating for the reader – since this is defined in the Supplemental Materials the supplemental should be referenced here where μ is first introduced (similar to elsewhere in the main text).
- Regarding the above definition of μ in the Supplemental Materials, it would benefit to define the effect matrix for the Gaussian case (as opposed to the LG + G case only).
- There is some inconsistent use of notation in a few instances. For example, the effect matrices are labeled $M(x, y)$ in the main text, while $\hat{M}_F(x, y)$ is used in the supplemental, along with the issues with Fig. 2 and its caption highlighted above. The authors should use consistent notation throughout.
- Regarding the LG + G many-body entangled experiment, the spatial asymmetry between the atoms in the LG or G state is clear from the images, but the mention in the text is subtle. It would benefit the reader to include a few more sentences addressing this aspect (which the caption of Fig. 4 in the Supplemental Materials refers to).
- The figure captions in the Supplemental Materials are centre aligned, these should be left aligned.
- In general the formatting and presentation of the text leaves something to be desired for Nature Communications, given the several issues outlined above. The authors should carefully read through for consistent use of notation on submitting the final revision.

Referee 3 Comments	AUTHOR RESPONSE:
In Figure 2, it would be useful to label '0' on the horizontal axes and also increase the tick mark line widths to be more visible, as it isn't immediately obvious from the plots that $p(Q)$ ranges to negative values. There is also a font mismatch between the axes labels and the latex formatted figure caption and text.	Main text Fig. 2: modified as suggested. The figures and the text now use $p(Q)$ for the arrow of time distribution, and in general scalar variables have been formatted with italics, as per Nature Communications guidelines.
Also in Fig. 2, and the caption of Fig. 2, there is a lower case 'p' used for P (Q) which also mismatches the use in the text.	Modified.
It appeared to me that the lack of a definition of μ around Eqn. 4 is a bit frustrating for the reader –since this is defined in the Supplemental Materials the supplemental should be referenced here where μ is first introduced (similar to elsewhere in the main text).	Main text: μ appears in Eqn. 7 in this version. A reference to Supplementary Note 3 has now been added here, as well as a definition Eq. 8
Regarding the above definition of μ in the Supplemental Materials, it would benefit to define the effect matrix for the Gaussian case (as opposed to the LG + G case only).	Main Text: The main text now defines the effect matrix for the Gaussian case, along with the definition of μ in Eq. 8. Supplementary: The form of the effect matrix for the GG case has been added to Supplementary Note 3.
There is some inconsistent use of notation in a few instances. For example, the effect matrices are labeled $M(x, y)$ in the main text, while $\hat{M}_F(x, y)$ is used in the supplemental, along with the issues with Fig. 2 and its caption highlighted above. The authors should use consistent notation throughout.	Thank you, notation has been made consistent, with hats used for operators.
Regarding the LG + G many-body entangled experiment, the spatial asymmetry between the atoms in the LG or G state is clear from the images, but the mention in the text is subtle. It would benefit the reader to include a few more sentences addressing this aspect (which the caption of Fig. 4 in the Supplemental Materials refers to).	Main text Fig.1: Added a cartoon for the LG and G spatial modes in Fig.1, and a mention of the resulting spatial asymmetry in the absorption image here. Main text: Added a brief

	discussion attributing the asymmetry of the form of the fluctuation theorem to the singularity at the center of the LG mode, just before the Discussion section.
The figure captions in the Supplemental Materials are centre aligned, these should be left aligned.	Thank you, left aligned captions
In general the formatting and presentation of the text leaves something to be desired for Nature Communications, given the several issues outlined above. The authors should carefully read through for consistent use of notation on submitting the final revision.	Thank you, we have now made several changes where notational inconsistencies were noted.